# Saddles as rotational locks within shape-assisted self-assembled nanosheets

Joseph F. Woods [1], Lucía Gallego [1], Amira Maisch[1], Dominik Renggli[1], Corrado Cuocci[2], Olivier Blacque [1], Gunther Steinfeld[3], Andres Kaech[4], Bernhard Spingler [1], Andreas Vargas Jentzsch [5] & Michel Rickhaus [1] ✉

Two-dimensional (2D) materials are a key target for many applications in the modern day. Self-assembly is one approach that can bring us closer to this goal, which usually relies upon strong, directional interactions instead of covalent bonds. Control over less directional forces is more challenging and usually does not result in as well-defined materials. Explicitly incorporating topography into the design as a guiding effect to enhance the interacting forces can help to form highly ordered structures. Herein, we show the process of shape-assisted self-assembly to be consistent across a range of derivatives that highlights the restriction of rotational motion and is verified using a diverse combination of solid state analyses. A molecular curvature governed angle distribution nurtures monomers into loose columns that then arrange to form 2D structures with long-range order observed in both crystalline and soft materials. These features strengthen the idea that shape becomes an important design principle leading towards precise molecular self-assembly and the inception of new materials.

Designing and producing thin two-dimensional (2D) materials remains a challenging prospect for scientists to this day. The isolation of graphene in 2004 represented a landmark achievement in science[1,2], showing that a single layer of carbon atoms has the capability to act as an electrical conductor for devices and finds use in sensing applications whilst maintaining a very lightweight compared to alternatives[3–5]. Such properties are largely due to the geometry of the material and its composites. The charge conductivity within a graphene layer relies on its planarity and ability to pass electrons over large distances without restriction. Engineering new materials with more tuneable properties than those which already exist is a key goal and one that may be fulfilled with assistance from supramolecular chemistry where covalent bonds can be replaced by weaker interactions between molecules to form polymers[6,7]. The non-covalent interaction of choice is hydrogen bonding[8–10], but other intermolecular interactions, and their combinations thereof, can yield equally strong results[11–14]. One aspect of design that can enhance these less classical modes of association is the shape of the monomer[15–17], which is often underappreciated in comparison to the mode of interaction (hydrogen bonding, π–π interactions).

Instead of using functional groups to drive assembly with directional interactions, one can also use the shape of a molecule to modify a planar π-surface such that preferred arrangements exist that lower the energy of the system. Molecular shape can, in simple cases with spherical symmetry, embody three types of classical Gaussian curvature, and their difficulty of synthesis typically increases when moving away from planar architectures. Flat molecules have no curvature and are most often seen in supramolecular assemblies due to their facile syntheses[18,19]. Bowl shapes, which possess a positive Gaussian curvature[20,21], are less common but are not as unheard of as negatively curved molecules[22–24]. Synthesis of positively curved molecules usually relies on replacing a central benzene motif with a five-membered ring whereas negative curvature is often adopted classically through a central seven-membered ring or non-classically with oppositely facing five-membered rings. This negative curvature is most often referred to

---

[1]Department of Chemistry, University of Zurich, 8057 Zurich, Switzerland. [2]Institute of Crystallography, CNR, Via Amendola, 122/O, 70126 Bari, Italy. [3]ELDICO Scientific AG, 5234 Villigen, Switzerland. [4]Center for Microscopy and Image Analysis, University of Zurich, 8057 Zurich, Switzerland. [5]SAMS Research Group, University of Strasbourg, Institut Charles Sadron, CNRS, 67200 Strasbourg, France. ✉e-mail: michel.rickhaus@chem.uzh.ch

as a saddle shape and incorporation of such topography into a molecule generates a convenient topographical profile for the enhancement of weak interactions between surfaces[15]. The idea that a saddle can geometrically restrict the translational and rotational components of motion between building blocks can play a large part in strengthening the existing interactions and is, in such cases, key to fostering a greater degree of order when compared to topographies without such curvature. Specifically, if the direction of supramolecular polymerization is constrained, then this will result in an increased order within the overall structure and provide a crucial handle to fine-tune its assembled structure.

We recently reported this effect of shape-assisted self-assembly (SASA) with heteroaromatic macrocyclic compounds that were functionalized at the periphery with alkyl chains[17]. Weak association between π-surfaces of the core of carpyridine molecules allows for 1D columnar growth, which is then further stabilized and enriched by van der Waals interactions of the aliphatic sidechains. Propagation in a second direction is then allowed through lateral contacts of these sidechains such that molecularly thin 2D nanosheets are obtained. However, because the stability of the secondary structure (i.e. 2D sheet) largely dominates the assembly process, it was not possible to fully elucidate the precise mechanism of self-assembly with these examples. Although it was established that a delicate equilibrium among the interactions is critical for nanosheet formation, it is so far uncertain as to what precise role the topography of the monomer plays regarding the mechanism of self-assembly.

Whilst self-assembly, in general, can be considered a "delicate equilibrium", the use of stronger and more concerted interactions has, in the past, allowed for consistent analysis of supramolecular polymerizations and the rational conception of higher levels of complexity[25,26]. Our reported example on SASA using carpyridines[17] exploits molecular shape to influence various weak interactions, in particular van der Waals. As such, it becomes apparent that variations to the traditionally neglected alkyl chains will have a more dramatic influence in SASA compared to hydrogen-bonded supramolecular polymers. Previously, we have developed further carpyridine analogs to examine whether the SASA process is one that holds true with variation in length of the sidechain.

Whilst dodecyl chains were too long and the absence of sidechain was insufficient to provide evidence of self-assembly, we can now report that the intermediate range is as fascinating as expected for such a rich interplay of forces. While all carpyridine derivatives between methyl and decyl were observed to form 2D nanosheets with different thicknesses, the morphology, and size of these nanosheets varied greatly depending on sidechain length. However, through examination of the self-assemblies at a molecular level, multiple methods of solid-state analysis have shown 2D layers to exist within the assembled structures, which in turn, contain loosely held columns of carpyridine monomers. This structural factor is retained across the carpyridine series and is one we would attribute to the molecular topography. The analysis of torsion angles and rotational restriction between individual components of these columns demonstrates the assistance effect that a saddle topography can have in the assembly process. This results in columns with long-range order but no overall twist, which is rarely observed in other columnar assemblies with planar or bowl-shaped topographies[18,27,28] (Fig. 1). To our knowledge, this approach is one of the first to investigate the polymorphism that evolves from crystalline to soft materials using an array of diffractive techniques to discern the exact role of the molecule within the assembly.

The curvature-induced restriction of rotation is consistent across derivatives and is a determining factor in the observed supramolecular ordering. A unique observation can also be made that as we progress from shorter sidechains with largely crystalline assembly modes, we can clearly observe the transition to materials of softer nature as the chain length increases[29–32]. The transition and co-existence between the two can be established as we observe differences between diffraction techniques that would lend a hand in the structural deduction of the assembly[33–35]. The role of shape is equally important across carpyridine derivatives, irrespective of the properties of the sheets, and showcases how a saddle topography can enhance ordering processes, particularly with respect to rotational freedom.

## Results

To investigate the extent to which the sidechains affect the assembly, a series of carpyridine derivatives with a variation in alkyl chain length

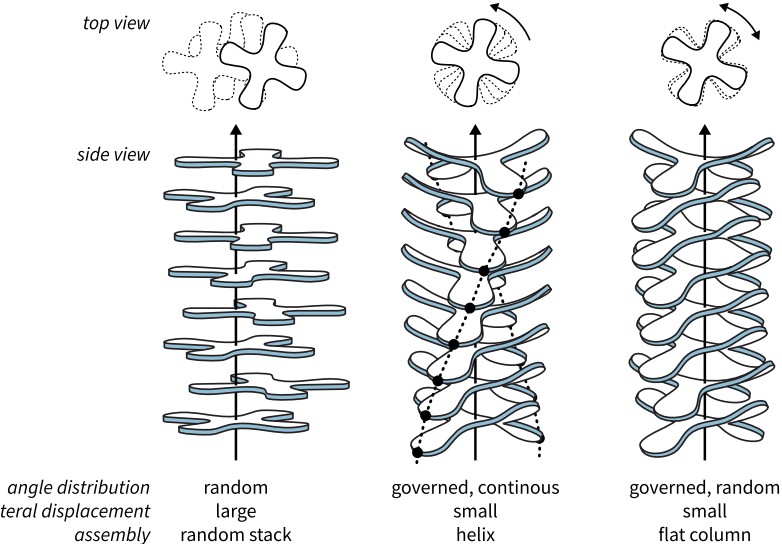

**Saddles as rotational locks under weak forces**

Without the assistance of shape there is random organization in the absence of strong, directional forces. Adding such interactions (H-bonding) can lead to order but monomers keep the same orientation because of this. Topography and weak interactions lead to angle governance arising from shape

top view

side view

| angle distribution | random | governed, continous | governed, random |
| lateral displacement | large | small | small |
| assembly | random stack | helix | flat column |

**Scope**

Retaining the macrocyclic core and varying the peripheral sidechain leads to different degrees of self-assembly

2H-Car-R

-C1 — Me     -H — H

-C2 — Me

-C3 — Me                   chain elongation

-C4 — Me

⋮

-C10 — Me

**Fig. 1 | Assembly principle and carpyridine scope.** Using saddles as rotational locks, left, allows for generation of 1D stacks with no overall twist that can then lead to materials of higher dimensionality and, right, the scope of carpyridine derivatives that only vary in the length of alkyl chain.

were prepared ranging between methyl and decyl (**2H-Car-CX**, where X corresponds to the number of carbons in the sidechain). The synthesis of these compounds follows similar synthetic procedures to those previously reported[17]. The full characterization of the carpyridines and their intermediates is described in the Supplementary Information.

There is a notable difference in the solubility of the carpyridines in organic solvent depending upon chain length. The methyl analog, **2H-Car-C1**, is less soluble than the naked derivative (**2H-Car-H**) despite its substitution although solubility increases as the chain length is extended through the series. All carpyridines adopt the same spectral profiles in UV-vis absorption and fluorescence spectroscopy (Supplementary Fig. 62) and exhibit modest quantum yields of fluorescence ($\Phi_f = 41$–$56\%$). The similarities of these molecules further corroborated our previously reported observations that only minor changes can be observed using these spectroscopic techniques at their relevant concentrations.

To examine the extent of interaction in solution, variable temperature $^1$H NMR spectra of selected carpyridines were performed in toluene-$d_8$ (all 9.2 mM, Supplementary Figs. 57–61). All species show similar behavior to that of **2H-Car-C6** with increasing broadness as the temperature is lowered but also reveal multiple environments at higher temperatures. The same coalescence of pyridine signals was observed although at different temperatures over the series. Carpyridines with sidechains shorter than pentyl coalesce at 223 K but the same phenomenon occurs at 243 K with **2H-Car-C5**. This then increases to 253 K in the cases of **2H-Car-C6** and **2H-Car-C10**, which hints at a difference in behavior between shorter and longer chained derivatives.

To investigate the assembled structures, we first performed an initial study with transmission electron microscopy (TEM) for all carpyridines (Fig. 2). Solutions of 1 mM concentration were prepared under identical conditions for each sample in toluene before dropcasting onto C/Cu TEM grids. Large, thin flake-like sheets with lengths up to 10 μm were observed for **2H-Car-C1** yet the addition of a methylene unit to the sidechain showed that **2H-Car-C2** assembles into equally large but square nanosheets, which consist of multiple layers. Atomic force microscopy (AFM) of **2H-Car-C2** (Supplementary Fig. 66) demonstrated the multi-layered nature of the sheets with observation of 1.5 nm offsets accounting for individual layers. Less well-defined assemblies were seen with the propyl sidechain but an increase in assembly size was found in multi-layered and curved **2H-Car-C4** sheets with lengths up to 40 μm. The larger dimensions and thicknesses were retained by **2H-Car-C5** alongside some extent of layering, although these sheets tended to showcase a sharpness at assembly edges, yielding more defined prismatic bodies than the reported **2H-Car-C6** assemblies but without the single molecular thicknesses. Expanding the length of sidechain above the hexyl functionality showed a decrease in assembly constitution in terms of the distinct, sharp boundaries and sheet size observed between **2H-Car-C7**, which largely resembled the structures observed with **2H-Car-C6**, and **2H-Car-C10** that indicated almost no retention of supramolecular ordering suggesting that the interactions between sidechains overcome the balance around this chain length.

Selected area electron diffraction (SAED) of the carpyridine TEM samples was then conducted, where diffuse halos were seen in several cases. **2H-Car-C6**, **2H-Car-C7**, and **2H-Car-C8** sheets all contained the observed principal distance of ~4 Å that can be ascribed to π–π distances between carpyridine cores. The SAED patterns of shorter-chained derivatives did not contain this specific diffraction, however, highly ordered arrays of discrete diffraction spots were observed in the cases of **2H-Car-C1**, **2H-Car-C2**, **2H-Car-C4**, and **2H-Car-C5** instead. In fact, remeasuring SAED on new samples of **2H-Car-C6** allowed us to also observe similar diffraction patterns which we had not observed previously (Supplementary Fig. 65). The analysis of these diffraction patterns yielded characteristic distances that represent two of the

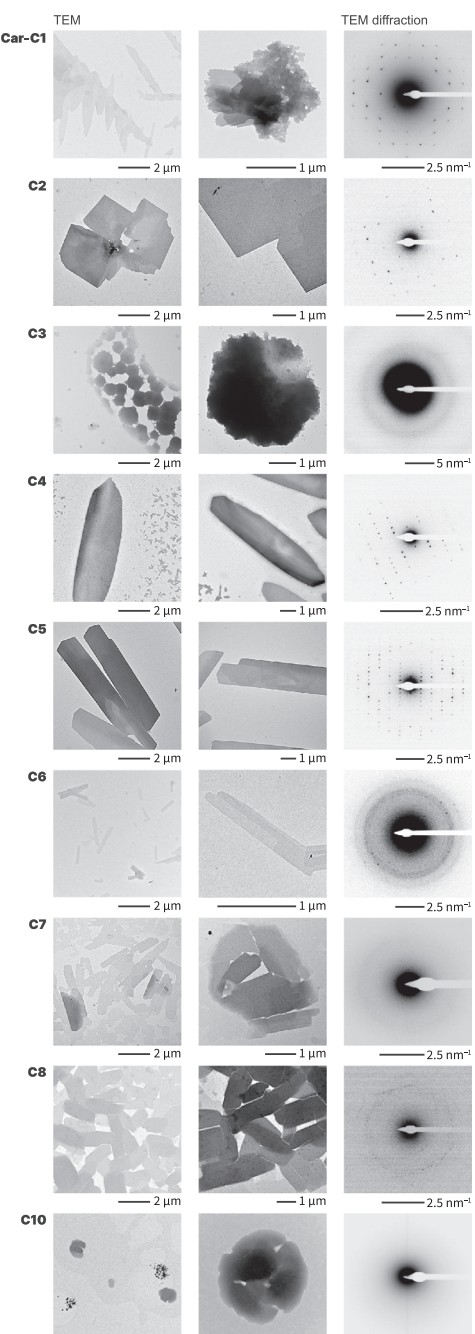

**TEM and TEM diffraction**

◄◄ Dropcasting 1 mM solutions of 2H-Car-R onto C/Cu TEM grids provided 2D nanosheets in all cases but with different morphologies and characteristics.

◄ Each assembly diffracts to different extents. C1–C5 generally diffract well and yield discrete spots (except for C3, see text) that could be related to unit cell parameters whereas C6 and above showed diffuse halos that had a characteristic distance of ~4 Å until no diffraction was observed with C10.

**Fig. 2 | TEM images and SAED patterns of carpyridines.** 2D sheets and their diffractions were observed for all species except for **2H-Car-C10**, which did not diffract.

three dimensions within the repeating unit cell of the assembled structure, with the missing dimension being perpendicular to the surface (Supplementary Figs. 64 and 65).

The remarkable order and diffracting power of our self-assembled structures prompted us to investigate the molecular arrangement in single crystals that could be obtained from toluene/methanol mixtures for **2H-Car-C2** to **2H-Car-C6**. An increase in sidechain length from **2H-Car-C2** to **2H-Car-C6** is the only perturbation to the macrocyclic core and for chains above that of ethyl, a consistent trend can be observed across the series. Carpyridine monomers pack into zig-zagged antiparallel columnar arrays that repeat across the structure (Fig. 3). An additional striking feature observed within the single crystal structures was the retention of the orientation of carpyridine cores relative to one

## From crystaline states...

For chain lengths C2 to C4, the solid state structures match that of the assembly on surface exactly. C3 and above show columnar, antiparallel 2D arrangements that stack as layers whereas C2 exists in a 2D network of dimers.

## ...to soft matter

From C5 on, the solid state structure (shown) no longer matches that of the on-surface assembly exactly, yet both retain columnar 2D packing.

### 2H-Car-C2
$P$ = 1.32 nm | $U/P$ = 5
$P/U$ = 0.26 nm

### 2H-Car-C3
$P$ = 1.32 nm | $U/P$ = 5
$P/U$ = 0.26 nm

### 2H-Car-C4
$P$ = 1.30 nm | $U/P$ = 5
$P/U$ = 0.26 nm

### 2H-Car-C5
$P$ = 1.22 nm | $U/P$ = 5
$P/U$ = 0.24 nm

### 2H-Car-C6
$P$ = 1.74 nm | $U/P$ = 7
$P/U$ = 0.25 nm

▲ Single crystals of carpyridines were grown from toluene/methanol mixtures. The unit cell was expanded to reveal loose columnar arrays of antiparallel stacks for C3 and longer sidechains. Solid state structures of C5 and C6 overall deviate from those of the assembly, but columnar packing can still be observed in the single crystal. $P$ = column period, $U$ = unit.

▲ Similar behavior is exhibited between C3 and C5, where three molecules span the column width before an increase to four molecules at C6. However, the columnar proportions per carpyridine unit are consistent across all columnar assemblies. The properties of the columns are compared with each other and a known[28] H-bonded curved π-assembly.

**Fig. 3 | Solid state structure of carpyridine single crystals.** X-ray crystallography of carpyridines grown from toluene/methanol revealed loose columns with similar properties to exist across derivatives with chains lengths of three carbons and above whereas **2H-Car-C2** exists as a defined 2D network of dimers. The lower panels show the behavior of the columns with respect to the carbon sidechain length, which includes the dimensions per unit carpyridine and the extent of rotation between monomers. Sidechains are omitted for clarity and distances are specified in nm.

another. Fluctuation of the carpyridine-carpyridine torsion angle within the columns is limited to a maximum of 8.5° in the case of **2H-Car-C4**, with all other crystal structures indicating adjacent rotational displacements between monomers to be smaller than this across the other derivatives.

Upon cross-reference of the SAED patterns with the X-ray single crystal structures of the respective carpyridines, the unit cell dimensions of **2H-Car-C2** and **2H-Car-C4** were found to match those of the SAED pattern. The crystal structure of **2H-Car-C3** contained a high level of disorder in the sidechains and we consider this to be the reason for a lack of diffraction under TEM. Alignment of the crystal structures along the axis absent from the SAED and generation of the molecular packing allowed for visualization of the supramolecular assemblies upon the surface (Fig. 3) given the matching diffractions. This was then further corroborated through micro-electron diffraction (μED) of the structures observed by TEM that resulted from the self-assembly process of **2H-Car-C4**. However, the overlap between the crystal packing and the self-assembled structures diverted at the point of **2H-Car-C5**, indicating that both crystallization and supramolecular assembly result in different molecular arrangements or polymorphs (Fig. 4). This is important because it reveals the exact boundary at which the carpyridine core provides order locally but does not fully determine the long-range order that results from the additional input

from the alkyl chains: the ensemble of molecules behaves as soft material.

Multi-layer thick carpyridine nanosheets of **2H-Car-C5** and **2H-Car-C6** provided diffraction patterns using μED that were in agreement with the obtained SAED patterns. Due to the weakly diffracting nature of these longer chained carpyridine nanosheets, a precise solution could not be found in the case of **2H-Car-C6**, however, using the molecular geometry from the crystal structure of **2H-Car-C5** and mapping this to the μED diffractions, a best-fit model showed that a similar antiparallel columnar arrangement exists within the assembly (Supplementary Fig. 77). The 1.5 Å μED dataset of **2H-Car-C5** was solved by simulated annealing (see Methods section for more details), which makes this, to the best of our knowledge, one of the first examples[36–38] of such a low-resolution small molecule μED structure being solved.

The columnar thicknesses are increased within the **2H-Car-C5** assembly compared to the crystal structure due to four carpyridine molecules contributing to the width of a single column as opposed to three in the crystal and columns interpenetrate one another to a lesser extent. Crucially, as seen in the crystal structures, the relative orientation of carpyridine cores appears to be preserved in the assembly such that there is very limited rotational freedom within the column despite the greater translational slippage. Although ultimately no proposed structure could be derived for **2H-Car-C6**, the unit cell

## A direct comparison between C4 (crystalline) and C5 (soft)

For C4 the observed unit cell of the crystal and on surface match, while C5 deviates. The on-surface methods SAED and μED agree on the same unit cell. Notably, C5 shows multiple phase transitions while the one carbon shorter C4 does not.

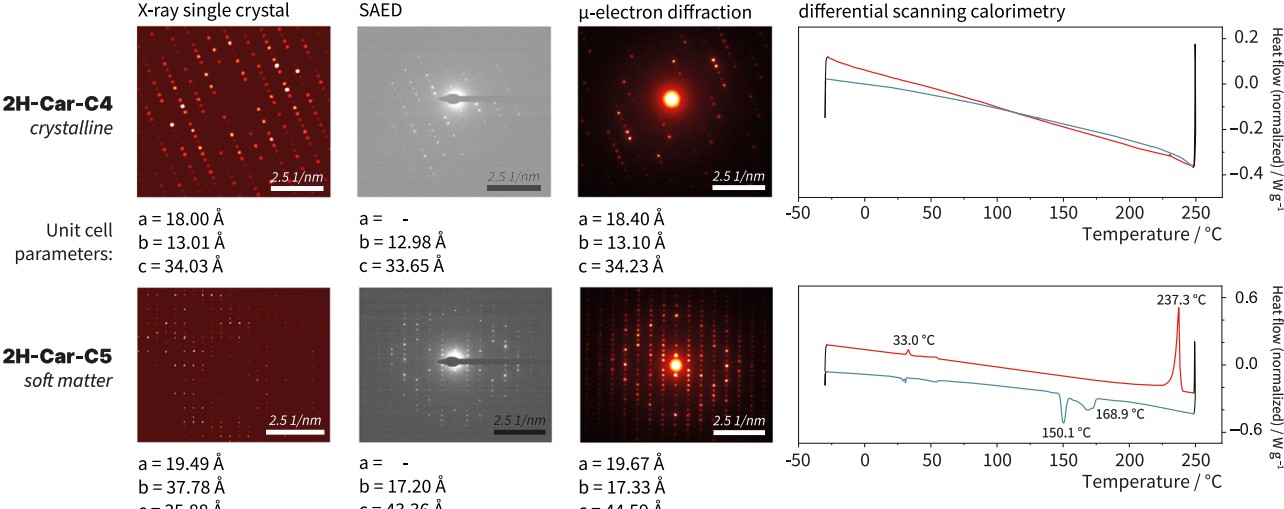

**Fig. 4 | Comparison between crystalline and soft carpyridines.** Behavioral changes are detected moving from **2H-Car-C4** to **2H-Car-C5** in the solid state with crystal structures no longer matching the assembly on surface (SAED and μED). Differences are also seen with DSC as phase transitions are observed with **2H-Car-C5** but not with **2H-Car-C4**.

dimensions from μED revealed key dimensions with lengths matching those found in the SAED pattern. In particular, the unknown cell length of 20.96 Å was determined, which matches well with the observed thickness of a molecular layer measured with AFM (2 nm). In both of these cases, the weakly diffracting nature and impossibility of finding an exact solution for **2H-Car-C6** support the observation of a transition towards soft matter from crystalline material.

These findings are corroborated by thermal analysis with differential scanning calorimetry (DSC), which revealed behavior reflecting that of a crystalline material for the shorter-chained carpyridines (Supplementary Fig. 68). No distinct thermal events could be observed between −30 °C and 300 °C for **2H-Car-H** to **2H-Car-C4**. On the other hand, phase transitions before melting were seen with **2H-Car-C5**, and indeed with all derivatives with longer sidechains, confirming that a different behavior is exhibited by these systems. When cooling down from the isotropic melt, exotherms were also seen in these systems unlike the derivatives with shorter sidechains. Based on the previously observed birefringence for **2H-Car-C6**[17], these phase transitions for the carpyridines with chain length above **C5** can be attributed to at least one partially (liquid) ordered state. The observations seen with polarized optical microscopy of **2H-Car-C6** are consistent with liquid crystalline material although this is yet to be confirmed by X-ray scattering.

## Discussion

The generality and limitations of the assembly of carpyridines into 2D supramolecular sheets were investigated, which has resulted in a noteworthy example of self-assembly as these units do not make use of conventional approaches based on strong, highly directional driving interactions such as hydrogen bonding yet achieve an equally high level of supramolecular order. While the weaker nature of the assembling interactions implies that variations to the sidechains will have a greater impact here than for classical examples, both the large variation of assembled motifs and the contrasting consistency in the manner that order is imposed on the overall system are remarkable.

Furthermore, the systematic investigation of alkyl chain length provides insight as to how control over the assembly may be had, as well as the challenges associated with the use of weaker interactions

through the SASA principle. Whilst one component of attraction between monomers arises from the macrocyclic π-cores, tempering the extent of the van der Waals interactions at the periphery has a large impact upon the extent of self-assembly[11,39]. Although linear columns were not directly observed in the resolved structures, repetitive intercalating columnar arrays can be seen across the series that restrict the orientation of carpyridine molecules.

Two relevant questions need to be addressed: (i) can the shape of the carpyridine motif determine the formation of layered sheets and (ii) how is this order governance translated to the assembly of soft materials? To establish the existence of these columnar stacks, X-ray crystallography was deemed the most useful technique. However, as the crystallization conditions (in this case ~5 mM from toluene/methanol) do not necessarily match the conditions of self-assembly, we decided to exploit both SAED and μED structural elucidation to investigate the structural parameters of the ultrathin 2D sheets observed. Together, these three techniques allowed us to gain unique insights into the way carpyridines instigate order, to observe the transition from a solid and mostly crystalline material to a (dynamic) soft material, and to establish a landmark example on the competition of shape, π-π interactions, and alkyl chain association. All three methods of analysis were in agreement for the unit cell parameters of **2H-Car-C2** and **2H-Car-C4**. However, different polymorphs with different cell parameters were obtained from solution-grown crystals and measured by X-ray crystallography that allowed for comparison with the assemblies formed through evaporation of dilute solutions on grid surfaces and the data collected with SAED and μED for **2H-Car-C5** and **2H-Car-C6**. Beyond **2H-Car-C6** only the diffuse and weak characteristic distances for the carpyridine stacking and alignment were observed as halos on the SAED measurements.

The morphology of the nanosheets can be attributed to the molecular packing directly. For example, as **2H-Car-C2** is composed of dimeric units and these units have an offset approximately perpendicular to one another, it is unsurprising that a square sheet is observed in this case. The layered structure observed in TEM and AFM arises from the packing into 2D layers spanning a 1.5 nm length of the molecule (see the 1.57 nm distance observed in the middle panel of Fig. 3). This packing mode does not hold true in **2H-Car-C3** or **2H-Car-C4**

and is overcome by preference for zig-zagged column formation that results in vertical slipping of columns and accounts for the lack of a defined edge observed in these assemblies with shorter sidechains. Although the crystal structures for **2H-Car-C5** or **2H-Car-C6** do not match the assembled structure, the same behavior can be observed in the single crystals and, indeed, in the derived assembled structure for **2H-Car-C5**. We anticipate that the perceived lack of growth in the third dimension is due to the weaker van der Waals interactions between alkyl chains. Propagation of the assembly in two dimensions is favored because of the eclipsing aryl groups but access to 3D structures is inhibited due to layer segregation.

Orientation of carpyridine cores across all crystal structures is notable due to the lack of rotation around one another. While some translational displacement is observed in one dimension, this is not true for rotations. The natural consequence of the formation of non-twisting stacks (Fig. 3 lower panel) results in the overall layered structure that is observed. This conclusion can be directly extrapolated to soft materials based on the carpyridine motif, and self-assembling stacking saddle monomers in general. The driving force for the association originates mainly from the π−π interactions between the aryl cores of the molecules combined with a van der Waals contribution from the alkyl sidechains as no other interaction has been identified in any of the molecules described here. It is also apparent that this interaction is enhanced by the shape the monomer possesses. Due to the saddle topography of the carpyridine, the barrier to rotation is heightened resulting in a significant restriction of rotation. Although translation in one direction is permitted to retain the columnar arrays, large rotations appear to be forbidden and such motion would result in too much disorder, prompting collapse of the assembly. The flexibility of the monomers towards translational displacement results in the different polymorphs observed, and we propose that it also accounts for the molecular arrangement within the **2H-Car-C6** assembly. Given the expansion in columnar width observed in the case of **2H-Car-C5** when moving from crystal structure to assembly, examination of the unit cell parameters for the **2H-Car-C6** single crystal and assembly would suggest a similar phenomenon. The columnar width likely relies on a contribution from four separate carpyridine molecules in both the crystal state and self-assembled one, hence the smaller difference between cell parameters (Supplementary Table 12).

With these findings, we believe we have strengthened the argument for SASA with the simplicity of carpyridine molecules. Modification of sidechain length by one methylene at a time has resulted in great variation and, to some degree, control of self-assembly[11,39]. The formation of nanosheets with sidechains of short lengths shows that the assembly is not fully reliant upon solvophobicity but more so on the strength of van der Waals interactions to compliment column formation between the aryl cores. Progression from crystalline material to soft matter has illustrated the continuity of the SASA approach, demonstrating that the scope is not limited to singular examples while also providing a unique example on the intrinsic challenges associated. The argument of shape being able to contribute to self-assembly has been further developed through the demonstration of a consistent restriction of rotation due to the molecular curvature, which is highlighted by the ability of carpyridines to consistently form 2D materials. Understanding this use of topography as a design concept will expand the limit of SASA to bring about new materials with alternative morphologies to nanosheets. The implications of these findings further push the appeal of using a topography-based concept towards the design of new materials, and in particular 2D supramolecular platforms.

## Methods
### General information
All reagents and solvents were purchased from Avantor, Chemie Brunschwig AG, Sigma-Aldrich, or Thermo Fisher and used without further purification. Dry solvents were obtained using a solvent purification system (Pure Solv PS-MD-4EN, Innovative Technology Inc.) equipped with alumina drying columns under argon. Reaction control was performed using analytical thin layer chromatography (TLC) on aluminum sheets coated with silica gel 60 F254 (Merck) or gas chromatography-mass spectrometry (GCMS) (Shimadzu Gas Chromatograph GC-2010 Plus, GCMS-QP2010 SE). Visualization of TLC plates was achieved using UV light at 254 or 366 nm. Flash column chromatography (FC) was performed using $SiO_2$ (60 Å, 230−400 mesh, particle size 0.063−0.200 mm). Preparative gel permeation chromatography (GPC) was carried out on a Shimadzu recycling GPC system equipped with an LC-20AR prominence liquid chromatograph pump, an SPDM40 photodiode array detector, a DGU-403 degassing and a CBM-40 system controller using two ReproGel 500 GPC columns (5 μm, 20×600 mm) with chloroform as the eluent passing through at a rate of 3.5 mL per minute.

All NMR spectra were recorded on AV2-400 or AV2-500 MHz Bruker spectrometers at 298 K unless stated otherwise. Chemical shifts are given in ppm and the spectra are calibrated using the residual chloroform signals (7.26 ppm for $^1$H NMR and 77.00 ppm for $^{13}$C NMR), the residual dimethylsulfoxide signals (2.50 ppm for $^1$H NMR and 39.52 ppm for $^{13}$C NMR), the residual tetrahydrofuran signals (3.58 ppm and 1.72 ppm for $^1$H NMR and 67.58 ppm and 25.38 ppm for $^{13}$C NMR) and the residual toluene signals (7.09, 7.00, 6.98 and 2.09 ppm for $^1$H NMR). Coupling constants $J$ are given in Hz and multiplicities are abbreviated as follows: s (singlet), d (doublet), t (triplet), m (multiplet), br (broad).

High-resolution mass spectra (HR-MS) were recorded by the Mass Spectrometric Service at the University of Zurich on a Dionex Ultimate 3000 UHPLC system (ThermoFischer Scientifics, Germering, Germany) connected to a QExactive MS with a heated ESI source (ThermoFisher Scientific, Bremen, Germany) or on a double-focusing (BE geometry) magnetic sector mass spectrometer DFS (ThermoFisher Scientific, Bremen, Germany) with a heated EI source.

UV-vis absorbance measurements were recorded at 298 K or at the given temperature with a Shimadzu UV-Visible Spectrophotometer UV-1900 Series using quartz 1 cm or 1 mm cuvettes.

Fluorescence measurements were carried out using a calibrated Edinburgh Instruments FS5 spectrofluorometer equipped with an SC-25 Temperature Controlled Holder TE-Cooled-Standard cell for emission spectra and an SC-30 Integrating Sphere cell for obtaining quantum yields. All solvents used for spectrophotometric analysis were of analytical grade.

DSC measurements were conducted using a Mettler Toledo DSC 3+ instrument. Approximately 1−2 mg of each sample were analyzed in 40 μl Al crucibles with pierced lids. A heating/cooling rate of 5 °C/min was used for a temperature range between −30 and 250 °C unless otherwise specified. The furnace was kept under inert atmosphere ($N_2$, 1 bar, 50 mL/min). The thermal properties were analyzed using the second and third heating cycles.

### Sample preparation
UV-vis absorbance measurements were carried out with ca. $10^{-5}$ M solutions in the solvent specified using a 1 cm quartz cuvette. Emission spectra and fluorescence quantum yields were measured on samples with an optical density of 0.06−0.10 using a 1 cm quartz cuvette.

The samples for electron microscopy were prepared at a typical concentration of 1 mM solution in toluene as follows: a 1 mM solution in toluene (20 to 100 μL) in a sealed vial was heated to about 100 °C for a few seconds and was allowed to regain room temperature. Throughout the process, the vials were kept at a uniform temperature and concentration by slow convexing movements. After 10−15 min at room temperature, 5 μL of the formed dispersion were deposited on a C/Cu grid. The solution was immediately blotted off the grid with a filter paper. After full evaporation of the solvent, the samples were used for data collection.

The samples for atomic force microscopy were prepared from the same solutions used for TEM imaging as described above. 5 μL of solution were deposited on a glass microscope slide and allowed to evaporate. After full evaporation of the solvent, the samples were used for AFM imaging.

## Microscopy

TEM diffraction and imaging was performed using an FEI Tecnai G2 Spirit microscope (Thermo Fisher Scientific, Eindhoven, The Netherlands) equipped with a LaB$_6$ cathode operated at 120 kV acceleration voltage and by using a side-mounted digital camera Gatan Orius 1000 (4k × 2.6k pixels, Gatan GmbH, Munich, Germany).

For diffraction, a selective area aperture of 685 nm diameter and an instrument camera length of 2.75 m was used. Diffraction patterns were calibrated using an evaporated Al standard and distances were compared to literature known values[40] of Al. Observed values for the aluminum standard were found to correspond to the literature value of 4.0494 Å when a calibration factor of 1.055 was applied and such a factor was applied to all measurements.

AFM was performed using an Asylum Research atomic force microscope (MFP-3D), which was used to measure the surface morphology in tapping mode. The probe used for the measurement was a HQ:NSC15/Al BS from MikroMasch. The Asylum Research built in software and Gwyddion (64 bit) were used to further analyze the AFM images.

## Electron diffraction measurements

All measurements were made on an ELDICO ED-1 electron diffractometer equipped with a LaB$_6$ source operating at 160 kV (λ = 0.02851 Å), producing a parallel beam of 750 nm in diameter[41]. The crystals were located and aligned by STEM imaging. Diffraction data were collected with a DECTRIS QUADRO® hybrid-pixel detector in steps of 0.5° per frame while the crystal was continuously rotated over a range of 60–150°. The software APEX4® by Bruker[42] was used to analyze the frames, to determine the unit cell constants and the integrated intensities. For the case of **2H-Car-C5**, diffraction spots were only observed until 1.5 Å resolution. Structure solution was attempted with the programs SHELXT[43], SHELXD[44] (both programs employ ab initio methods), Patsee[45], and Phaser[46] (within Phenix[47]) with the latter two using molecular replacement (MR). Eventually, the structure was solved with a real-space method based on the simulated annealing algorithm implemented in SIR2021, the updated version of the SIR2014 software[48]. The method involves the generation of a random sequence of trial structures starting from an appropriate 3D model, which is moved until a good match is attained between the calculated and experimental data through the evaluation of an appropriate cost function. For the structure solution, data up to 2.0 Å resolution were used and 1643 reflections were selected. The starting model was assembled from the molecule of **2H-Car-C5** derived from the crystal structure of a polymorph determined by X-ray methods. Three molecules of **2H-Car-C5** were positioned into the unit cell (Z′ = 3) assuming the space group P2$_1$/c. A total of 66 parameters were found by SIR2021 during the minimization process: three coordinates to describe the position of the center of mass, three describing the orientation, and 16 torsion angles to describe the conformation, for each molecule. The MPI implementation of the simulated annealing algorithm was run 100 times under a Linux virtual machine in a parallel calculation over 40 CPUs. The number of moves per step of temperature was increased to make the location of the global minimum more robust by using the directive niter 2000 in the SIR2021 input file. The total time required for the calculation was 7.3 days. The cost function was an R-factor that compares the experimental structure factors F$_o$ with the structure factors F$_c$ calculated by the current structure model: $CF = [\Sigma(F_o{}^2 - F_c{}^2)^2 / \Sigma(F_o{}^2)^2]^{1/2}$, where the summation is over the number of reflections in the experimental selected range. The best solution with the lowest cost function value was selected and successfully refined with ShelxL-2018[49] in Olex2[50] against the data at 1.5 Å obtaining R$_1$ values for strong data of ~27%, however, due to the low resolution, no improvement of the model could be observed.

## X-ray diffraction

Crystals were obtained from toluene and MeOH and were mounted on a cryo-loop and used for a low-temperature X-ray structure determination. All measurements were made on a Rigaku Oxford Diffraction XtaLAB Synergy diffractometer[51] with a Pilatus 200 K hybrid-pixel area detector using Cu Kα radiation (λ = 1.54184 Å) from a PhotonJet micro-focus X-ray source and an Oxford Cryosystems Cryostream 800 cooler. Data reduction was performed with CrysAlisPro[51]. The intensities were corrected for Lorentz and polarization effects, and a numerical absorption correction[52] was applied. The space group was uniquely determined by the systematic absences. Equivalent reflections were merged. The structure was solved by dual space methods using SHELXT-2018[43], which revealed the positions of all non-hydrogen atoms. Neutral atom scattering factors for non-hydrogen atoms were taken from Maslen, Fox, and O'Keefe[53], and the scattering factors for H-atoms were taken from Stewart, Davidson, and Simpson[54].

## Data availability

All data are available either in the main text, the supplementary materials, or by request of the corresponding author. The following .cif files are available on the CCDC database: **2H-Car-C2** (CCDC No. 2254105), **2H-Car-C3** (CCDC No. 2254084), **2H-Car-C4** (CCDC No. 2254085) and **2H-Car-C5** (CCDC No. 2254086).

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

## Acknowledgements

Imaging was performed with the support of the Center for Microscopy and Image Analysis, University of Zurich. The platform of polymer characterization at the ICS is acknowledged for DSC measurements. We also thank T. Fox for assistance with NMR measurements, T. Moehl for the use of AFM, the Mass Spectrometry Laboratory at the University of Zurich for MS measurements, and K. Gademann for generously hosting and supporting our research group. M.R. gratefully acknowledges funding from the Swiss National Science Foundation grant PZ00P2_180101 and J.F.W. is thankful for support from the University of Zurich Forschungskredit (FK-21-107).

## Author contributions

J.F.W., L.G., C.C., A.K., B.S., A.V.J., and M.R. conceived, analyzed, validated, and visualized the work herein presented. J.F.W., L.G., A.M., D.R., C.C., O.B., G.S., A.K., B.S., A.V.J., and M.R. carried out the investigation and developed the methodology. Funding was acquired by M.R. and

resources by B.S., A.V.J. and M.R. M.R. supervised the work. J.F.W., L.G., A.V.J., and M.R. wrote the manuscript.

## Competing interests

The authors declare no competing interests.
