## [Peer Review File · Nature Communications]

Saddles as rotational locks within shape-assisted self-assembled nanosheetsREVIEWER COMMENTS

Reviewer #1 (Remarks to the Author):

The manuscript reports a follow-up study of their previous Nature Communications paper published last year, which I enjoyed reading. Some of the results in the new manuscript were provided in the rebuttal letter for their previous paper (Peer Review File). As this fact indicates, the present study is closely related to the previous paper, and I suggest authors to combine these in a full paper and submit it to another journal. In my view, it contains an interesting concept, however, publication in Nature Communications cannot be recommended because it mainly reports detailed characterization and lacks the novelty. Systematic studies based on changing alkyl chain length is common in molecular self-assembly, liquid crystals, crystal engineering, and self-assembled monolayers, and not surprisingly, there should be structure-property relationship.

Reviewer #2 (Remarks to the Author):

In this communication, the authors present their findings on the consistent nature of shape-assisted self-assembly across various derivatives, emphasizing the restriction of rotational motion. The authors observe that a controlled distribution of angles, determined by the shape, facilitates the formation of loose columns from individual monomers. These columns subsequently organize themselves into 2D structures exhibiting long-range order. Notably, this phenomenon is observed in both crystalline and soft materials. These characteristics reinforce the notion that shape assumes a crucial role as a design principle, driving the precise self-assembly of molecules and enabling the emergence of novel materials. From my perspective, this paper signifies a significant advancement in knowledge that is likely to have a profound impact on the field. The approach employed by the authors is robust, and the quality of the data and its presentation is excellent. The conclusions drawn are well-articulated, backed by meticulously conducted experiments. In my view, no additional experiments are necessary. The paper is well-written, and the relevant literature has been thoroughly reviewed and incorporated. Given these considerations, I endorse the publication of this work in Nature Communications, with the highest priority.

Reviewer #3 (Remarks to the Author):

In this manuscript, Rickhaus et al. report the synthesis and in-depth characterisation of a systematic set of macrocycles, expanding on their previous study published in Nature Communications (DOI: 10.1038/s41467-022-31482-2) and providing further evidence for the role of molecular shape in their self-assembly.

From my point of view, this is a carefully conducted, methodologically sound study, which provides interesting insights that may guide the molecular design of other molecules with similar behaviour. Both the manuscript itself and the SI appear to have been prepared with great attention to detail. I can therefore recommend publication of this manuscript in Nature Communications after the following comments have been addressed:

1. The introduction of the term 'shape guided angle distribution' appears unnecessary to me and requires further justification and explanation. It is unclear how the introduction of this term facilitates the discussion of the self-assembly properties of the current or future molecules and what benefit it provides compared to available terminology (shape-assisted self-assembly, rotational restriction, ...).

2. According to the text in Figure 3, all systems show columnar, antiparallel 2D arrangements that stack in layers. With some imagination, this columnar arrangement can be seen for the molecules with longer alkyl chains, but I struggle to see this arrangement for 2H-Car-C2. The authors themselves describe the packing of 2H-Car-C2 as dimers arranged almost perpendicularly to one another at a later point in the manuscript, which appears to be the more accurate description.

3. Some further explanation for the formation of 2D structures instead of 3D structures should be provided.

4. It would be good to add a more detailed discussion/explanation of the panels at the bottom of Figure 3.

5. As the authors mention the good charge transport properties of some 2D materials in the

introduction, it may be interesting to compute the charge transfer integrals for their set of molecules. I would expect the charge transfer integrals to increase with increasing alkyl chain length / with more columnar arrangement.

Reply to Referee's Comments and Details of Revisions

Reviewer #1 (Remarks to the Author):

R1 The manuscript reports a follow-up study of their previous Nature Communications paper published last year, which I enjoyed reading.

Reply: We are glad to hear the reviewer enjoyed reading our previous work and thank them for their review of this work.

R1 Some of the results in the new manuscript were provided in the rebuttal letter for their previous paper (Peer Review File). As this fact indicates, the present study is closely related to the previous paper, and I suggest authors to combine these in a full paper and submit it to another journal. In my view, it contains an interesting concept, however, publication in Nature Communications cannot be recommended because it mainly reports detailed characterization and lacks the novelty. Systematic studies based on changing alkyl chain length is common in molecular self-assembly, liquid crystals, crystal engineering, and self-assembled monolayers, and not surprisingly, there should be structure-property relationship.

Reply: The rebuttal letter to the previous work did indeed include three of the seven novel assemblies reported in this new manuscript, but these were only provided as preliminary results as there was no conclusive knowledge of their structure at the time. The novelty of this manuscript details further examples of shape-assisted self-assembly and crucially, the structural characterisation down to the molecule of some of these assemblies using a multi-disciplined approach of solid-state analyses. The combination of methods that were used to elucidate the molecular arrangements is, to our knowledge, one of the few examples that exists for supramolecular materials. The combination of the structural parameters permitted the subsequent elaboration of a general picture describing the assembly process of these 2D materials, which was also not previously reported.

In particular, the method of structural deduction using the micro-electron diffraction data for the case of 2H-Car-C5 relied upon a simulated annealing technique that is highly novel.

As the reviewer has pointed out, this is a systematic study on a set of molecules, and it is perhaps unsurprising that there is a structure-property relationship. Yet, we would like to argue, that there is a critical need to engage in such systematic studies to fully characterize a supramolecular system. In our case, a general mode of assembly, which explains the (nano)microscopic arrangement could be found and can now be employed for the design of the next generation of carpyridine-based materials and related systems. Moreover, the study we report here is one of the few examples we are aware of in which the polymorphic evolution from crystalline to soft materials is studied down to the molecular arrangement level.

Based on the above, we believe we have both conceptual and methodological novelty. We have modified the text to better show the novelty of this particular manuscript with respect to the reported work.

Modifications to the manuscript:

- Page 3: "However, observations from solid state analyses have shown 2D layers within the assembled structures" is changed to "However, through examination of the self-assemblies at a molecular level, multiple methods of solid state analysis have shown 2D layers to exist within the assembled structures".
- Page 4: "We believe this approach is one of the first to investigate the polymorphism that evolves from crystalline to soft materials using an array of diffractive techniques to discern the exact role of the molecule within the assembly" has been added.

Reviewer #2 (Remarks to the Author):

R2 In this communication, the authors present their findings on the consistent nature of shape-assisted self-assembly across various derivatives, emphasizing the restriction of rotational motion. The authors observe that a controlled distribution of angles, determined by the shape, facilitates the formation of loose columns from individual monomers. These columns subsequently organize themselves into 2D structures exhibiting long-range order. Notably, this phenomenon is observed in both crystalline and soft materials. These characteristics reinforce the notion that shape assumes a crucial role as a design principle, driving the precise self-assembly of molecules and enabling the emergence of novel materials. From my perspective, this paper signifies a significant advancement in knowledge that is likely to have a profound impact on the field. The approach employed by the authors is robust, and the quality of the data and its presentation is excellent. The conclusions drawn are well-articulated, backed by meticulously conducted experiments. In my view, no additional experiments are necessary. The paper is well-written, and the relevant literature has been thoroughly reviewed and incorporated. Given these considerations, I endorse the publication of this work in Nature Communications, with the highest priority.

Reply: We thank the reviewer for their extremely supportive comments and strong endorsement of our work. We hope that this work can have a large impact upon the field as the reviewer predicts and that shape/curvature can be used as a key design principle for novel materials in the future. Nevertheless, we thank reviewers **R1** and **R3** for their additional comments to improve our manuscript further.

Reviewer #3 (Remarks to the Author):

R3 In this manuscript, Rickhaus et al. report the synthesis and in-depth characterisation of a systematic set of macrocycles, expanding on their previous study published in Nature Communications (DOI: 10.1038/s41467-022-31482-2) and providing further evidence for the role of molecular shape in their self-assembly.

From my point of view, this is a carefully conducted, methodologically sound study, which provides interesting insights that may guide the molecular design of other molecules with similar behaviour. Both the manuscript itself and the SI appear to have been prepared with great attention to detail. I can therefore recommend publication of this manuscript in Nature Communications after the following comments have been addressed:

Reply: We thank the reviewer for their supportive comments and are pleased that they recommend publication of our manuscript.

R3 1. The introduction of the term ‘shape guided angle distribution’ appears unnecessary to me and requires further justification and explanation. It is unclear how the introduction of this term facilitates the discussion of the self-assembly properties of the current or future molecules and what benefit it provides compared to available terminology (shape-assisted self-assembly, rotational restriction, ...).

Reply: Our original idea with the term “shape guided angle distribution” was to (re)introduce a term that can be used to describe molecular geometrical considerations that determine the likeliness of columnar stacks to, and to which extent, form helical structures. The interest of

the term would have been that it can be generally applied whereas these geometrical considerations are due to the curvature of the molecular system, the orientation of the hydrogen bonding groups, or any other molecular factors.

Nevertheless, we can see that it might be too early for such a generalization and that its use results in more open questions than it aids the general understanding of our message. Therefore, we have modified the manuscript, this time more clearly, referring to the “restriction of rotation”, which in our case specifically refers to a molecular-curvature restriction of rotation. We agree with the reviewer that this improves the readability of the manuscript and are thankful for the input.

Modifications to the manuscript:

- Abstract: “A shape governed angle distribution nurtures monomers into loose columns that then arrange to form 2D structures” is changed to “A **molecular curvature** governed angle distribution nurtures monomers into loose columns that then arrange to form 2D structures”.
- Page 4: “The SGAD phenomenon is consistent across derivatives and acts as a driving force for supramolecular ordering” is changed to “The **curvature-induced restriction of rotation** is consistent across derivatives and **is a determining factor in the observed supramolecular ordering**”.
- Page 13: “Due to the saddle topography of the carpyridine, the barrier to rotation is heightened resulting in SGAD” is changed to “Due to the saddle topography of the carpyridine, the barrier to rotation is heightened resulting in **a significant restriction of rotation**”.
- Page 14: “The argument of shape being able to contribute to self-assembly has been further developed through SGAD, highlighted by the ability of carpyridines to consistently form 2D materials” is changed to “The argument of shape being able to contribute to self-assembly has been further developed through **the demonstration of a consistent restriction of rotation due to the molecular curvature, which is** highlighted by the ability of carpyridines to consistently form 2D materials”.

R3 2. According to the text in Figure 3, all systems show columnar, antiparallel 2D arrangements that stack in layers. With some imagination, this columnar arrangement can be seen for the molecules with longer alkyl chains, but I struggle to see this arrangement for 2H-Car-C2. The authors themselves describe the packing of 2H-Car-C2 as dimers arranged almost perpendicularly to one another at a later point in the manuscript, which appears to be the more accurate description.

Reply: We have addressed this point in an updated Figure 3, stating that the columnar arrangements are only observed for cases of C3 and above and that C2 exists as a defined 2D network of dimers. This has been updated in the caption for the figure as well.

Modifications to the manuscript:

- Figure 3.
- Figure 3 caption: “X-ray crystallography of carpyridines grown from toluene/methanol revealed loose columns with similar properties to exist across derivatives” is changed to “X-ray crystallography of carpyridines grown from toluene/methanol revealed loose columns with similar properties to exist across derivatives **with chains lengths of three carbons and above whereas 2H-Car-C2 exists as a defined 2D network of dimers**”.

R3 3. Some further explanation for the formation of 2D structures instead of 3D structures should be provided.

Reply: We have added text to the manuscript explaining the preference for 2D sheets over 3D structures. This is due to the weaker vdW interactions between carpyridines that cause segregation of layers compared to the stronger interaction between π -surfaces which propagates the assembly in two dimensions.

Modifications to the manuscript:

- Page 13: “We anticipate that the perceived lack of growth in the third dimension is due to the weaker van der Waals interactions between alkyl chains. Propagation of the assembly in two dimensions is favored because of the eclipsing aryl groups but access to 3D structures is inhibited due to layer segregation” has been added.

R3 4. It would be good to add a more detailed discussion/explanation of the panels at the bottom of Figure 3.

Reply: We have added additional information within Figure 3 as an explanation of the lower panels that relate to the behaviour of carpyridine columns.

Modifications to the manuscript:

- Figure 3.
- Figure 3 caption: “The lower panels show the behaviour of the columns with respect to the carbon sidechain length, which includes the dimensions per unit carpyridine and the extent of rotation between monomers” has been added.

R3 5. As the authors mention the good charge transport properties of some 2D materials in the introduction, it may be interesting to compute the charge transfer integrals for their set of molecules. I would expect the charge transfer integrals to increase with increasing alkyl chain length / with more columnar arrangement.

Reply: We agree with the reviewer that this would be very interesting to explore. However, we see this investigation to be outside the scope of this current work. The computations would be an insightful addition, but we would like to combine this with experimental evidence. We have begun collaborations to investigate the charge transfer properties within the carpyridine assemblies and hope to report the findings soon.

REVIEWERS' COMMENTS

Reviewer #3 (Remarks to the Author):

My concerns have been addressed, but I would recommend replacing the term "shape governed angle distribution" also in Figure 1, given that it has been replaced in the rest of the manuscript.

Reply to Reviewer's Comments and Details of Revisions

Reviewer #3 (Remarks to the Author):

R3 My concerns have been addressed, but I would recommend replacing the term "shape governed angle distribution" also in Figure 1, given that it has been replaced in the rest of the manuscript.

Reply: We are pleased that we were able to address the concerns of the reviewer and thank them again for their helpful suggestions on improving the manuscript. We also thank them for spotting this section of the manuscript where we did not change the wording. We have made the changes.